# Single-Cell Transcriptomic Landscape of Cervical Cancer Cell Lines Before and After Chemoradiotherapy

**DOI:** 10.3390/cells15020115

**Published:** 2026-01-08

**Authors:** Dmitriy V. Semenov, Irina S. Tatarnikova, Anna S. Chesnokova, Vadim A. Talyshev, Marina A. Zenkova, Evgeniya B. Logashenko

**Affiliations:** Institute of Chemical Biology and Fundamental Medicine, Siberian Branch, Russian Academy of Sciences, Lavrentyev Avenue, 8, Novosibirsk 630090, Russia; istatarnikova@gmail.com (I.S.T.); a.chesnokova@alumni.nsu.ru (A.S.C.); talyshev.v@gmail.com (V.A.T.); marzen@1bio.ru (M.A.Z.)

**Keywords:** cervical cancer cell line, chemoradioresistance, single-cell RNA sequencing, differentially expressed genes, transcriptomic reprogramming, human papillomavirus

## Abstract

**Highlights:**

**What are the main findings?**
We established and characterized a unique pair of isogenic cervical cancer cell lines (AdMer35/AdMer43) from a patient with squamous cell carcinoma of the cervix before and after chemoradiotherapy, revealing that resistance is not caused by a single factor but by complex reorganization of cancer cells.Single-cell analysis identified key resistance hallmarks: transcriptional reprogramming towards embryonic stemness (HOX, POU5F1), a shift from fibrillar to non-fibrillar collagen expression, and activation of interferon/inflammatory pathways.

**What are the implications of the main findings?**
The study maps a remodeled tumor ecosystem with a metabolically reprogrammed, pro-tumorigenic senescent-like cell population and a progenitor-like cell reservoir, identifying them as critical therapeutic targets for eliminating resistant cell pools.The identified key pathways, Wnt, TGFβ, BMP, and the specific ligands EREG and SEMA3C, along with the metabolic vulnerabilities of the senescent compartment, provide a robust framework for future therapeutic exploration.

**Abstract:**

Cervical cancer remains a significant global health burden, with chemoradioresistance representing a major obstacle to successful treatment. To elucidate the mechanisms underlying this resistance, we established a unique pair of isogenic primary cervical cancer cell lines, AdMer35 and AdMer43, obtained from a patient with squamous cell carcinoma of the cervix before and after radiation therapy. The aim of our study was to characterize the transcriptomic and cellular heterogeneity of these cells. We conducted an in-depth comparative analysis using single-cell RNA sequencing. Analysis of this paired, patient-derived isogenic model suggests that chemoradioresistance can arise through coordinated multilevel cellular adaptations. Resistant AdMer43 cells demonstrated transcriptional reprogramming, with the upregulation of embryonic stemness factors (HOX, POU5F1, SOX2), a shift in extracellular matrix from fibrillar to non-fibrillar collagens, and activation of inflammatory pathways. We identified and characterized critical cell-state dynamics: resistant cells exhibited a remodeled ecosystem with a metabolically reprogrammed senescent-like cell population showing an enhanced pro-tumorigenic communication via EREG, SEMA3C, BMP, and WNT pathways. Furthermore, we identified a progenitor-like cell population with a minimal CNV burden, potentially serving as a reservoir for tumor persistence. These findings offer novel insights for developing targeted strategies to eliminate resistant cell pools and improve cervical cancer outcomes.

## 1. Introduction

Cervical cancer is the second most common malignancy and the second leading cause of cancer-related death among women of reproductive age worldwide [1].

Persistent infection with human papillomavirus (HPV), especially types 16 and 18, is the principal etiological factor for cervical cancer [2,3,4,5]. Despite the availability of effective screening programs and preventive HPV vaccination, the incidence and mortality rates of cervical cancer remain high and have even been increasing in low- and middle-income countries. In 2018, a total of 568,847 new cases and 311,365 cervical cancer-related deaths were reported globally. Notably, there is a trend towards an increase in the incidence among young women [6,7]. Cervical cancer comprises various histopathological subtypes (e.g., adenocarcinoma/adenosquamous carcinoma and squamous cell carcinoma), which are commonly treated with either definitive radiotherapy (RT) or concurrent chemoradiotherapy [8].

Radiotherapy directly induces DNA double-strand breaks in tumor cells, while reactive oxygen species (ROS) generated during RT further contribute to indirect DNA damage. On the cellular level, RT-induced DNA damage, particularly double-strand breaks, is generally lethal. However, when DNA repair capacity exceeds the rate of damage, tumor cells may survive RT, leading to the development of radioresistance [9]. In addition, accumulating evidence from preclinical studies shows that both radiation and chemotherapy can drive tumors and surrounding non-malignant cells into a state of senescence. These therapy-induced senescent cells may persist in tissues and, through secretion of pro-inflammatory and pro-tumorigenic factors known as the senescence-associated secretory phenotype (SASP), contribute to relapse, metastasis, and resistance to subsequent treatments [10]. In cervical cancer, radioresistance remains one of the principal barriers to effective therapy and durable disease control. Recent molecular studies indicate that this phenomenon arises from the interplay of genetic, epigenetic, and microenvironmental mechanisms. The ability of tumor cells to efficiently repair radiation-induced DNA damage allows them to survive treatment. They activate several repair systems, including homologous recombination, non-homologous end joining, and base excision repair. When key repair proteins such as RAD51 and PARP are overproduced, the cells gain a clear survival advantage after irradiation. Alterations in the p53 pathway suppress apoptosis and facilitate epithelial–mesenchymal transition (EMT), thereby promoting tumor persistence and metastatic potential after radiation exposure [11]. Epigenetic dysregulation, including aberrant DNA methylation and the altered expression of non-coding RNAs, such as miRNAs and lncRNAs, further modulates genes involved in DNA repair, cell cycle control, and stress adaptation. In parallel, the tumor microenvironment critically contributes to radioresistance. Hypoxia and activation of HIF-1α diminish oxygen-dependent DNA damage and drive metabolic reprogramming, while tumor-associated macrophages and cancer-associated fibroblasts secrete interleukin-6 and other growth-promoting cytokines that reinforce radioresistant phenotypes [12].

Overall, radioresistance in cervical cancer represents a multifactorial process involving coordinated molecular and microenvironmental adaptations that enable tumor cell survival, regrowth, and dissemination after RT. A more profound understanding of these mechanisms is essential for identifying predictive biomarkers and developing targeted radiosensitizing strategies to enhance treatment outcomes. Although major progress has been made in elucidating the molecular events underlying HPV-induced malignant transformation, many aspects of cervical cancer etiopathogenesis remain unclear [13]. Further studies are required to link these molecular alterations with clinical behavior and treatment response. Despite substantial advances in genomic and transcriptomic profiling, reliable biomarkers that can accurately predict therapy outcomes or patient survival are still lacking, highlighting the need for continued research to translate molecular discoveries into clinical applications.

Transcriptome profiling may help answer these questions, as it offers knowledge regarding gene structure and function, regulation of gene expression, and genomic plasticity. It may also reveal key changes in the biological processes that trigger human diseases, thereby offering new tools useful not only for understanding their underlying mechanisms but also for molecular diagnostics and clinical therapy [14]. Previous transcriptome-based studies of cervical cancer have revealed a cascade of molecular and cellular changes driving disease progression. Early cervical intraepithelial neoplasia is marked by activation of DNA replication, repair, and proliferative pathways, while progression to invasive carcinoma involves metabolic reprogramming with reduced mitochondrial and ribosomal activity, reflecting a shift toward glycolysis. In parallel, epithelial estrogen receptor alpha (ESR1) expression declines, whereas stromal ESR1 persists and associates with pro-inflammatory and angiogenic signaling, highlighting the importance of stroma–tumor interactions. Single-cell transcriptomic analyses further demonstrate the dynamic remodeling of the immune microenvironment, evolving from partial activation in precancerous lesions to immunosuppression in tumors, with signs of reactivation in metastatic lymph nodes. Collectively, these studies underscore how transcriptome profiling can illuminate the molecular, metabolic, and immunological transitions that shape cervical cancer development [15,16].

We established two novel primary human cervical cancer cell lines, AdMer35 and AdMer43, derived from the same patient before and after chemoradiotherapy [17]. We characterized these lines in terms of karyotype, morphology, proliferation, migration capacity, and sensitivity to radio- and chemotherapy. Our data revealed marked differences between the two lines, suggesting that therapy-induced changes may have led to phenotypic divergence.

The current data on the genes and pathways involved in the pathogenesis of cervical cancer and the mechanism of radioresistance do not provide a comprehensive understanding of the underlying molecular mechanism [18,19,20,21]. Therefore, in this study, we performed a deep transcriptome analysis of individual cells using a unique pair of cell lines, AdMer35 (one course of chemotherapy) and AdMer43 (derived from a recurrent tumor after chemoradiotherapy), from the same patient. This let us elucidate the mechanism of chemoradioresistance development and identify genes (pathways) activated/downregulated in these cells that were responsible for these changes. Our findings allowed us to directly compare the molecular landscape before and after therapeutic pressure.

## 2. Materials and Methods

### 2.1. Cell Lines

Cell lines AdMer35 and AdMer43 were obtained as described previously [17]. Briefly, both cell lines were derived from biopsy samples of a 22-year-old patient diagnosed with stage IIbN1M0 (IIIB) squamous cell carcinoma. The primary lines were developed from the samples obtained before (AdMer35) and after (AdMer43) chemoradiotherapy (contained in Supplementary Treatment Protocol of AdMer Patient). Cells were cultured in DMEM supplemented with 10% (*v*/*v*) heat-inactivated FBS and antibiotic–antimycotic mix (100 U/mL penicillin, 100 μg/mL streptomycin, 0.25 μg/mL amphotericin, and 50 μg/mL gentamicin) and, thereafter, complete DMEM. Cells were cultured at 37 °C in a humidified atmosphere containing 5% CO_2_.

### 2.2. Single-Cell RNA Sequencing

Cell counting and viability assays were carried out using 0.4% trypan blue (Thermo Fisher Scientific, Waltham, MA, USA) and a Luna II automated cell counter (Logos Biosystem, Dongan-gu, Anyang, Republic of Korea). Cell suspensions were diluted to 1000 cells/mL and placed on ice. Single-cell RNA sequencing was performed on the 10× Genomics Chromium Controller using the Single Cell 3′ Reagent Kit v3.1 (10× Genomics, Pleasanton, CA, USA). The number of cells in each sample did not exceed 10,000. cDNA amplification and library construction were carried out according to the manufacturer’s protocol. The concentration of cDNA libraries was measured using the dsDNA High Sensitivity kit on a Qubit 4.0 fluorometer (ThermoFisher Scientific, Waltham, MA, USA), followed by quality assessment with the High Sensitivity D1000 ScreenTape on a 4150 TapeStation (Agilent, Santa Clara, CA, USA). cDNA libraries were pooled, denatured, and sequenced using Genolab M (GeneMind, Shenzhen, Guangdong, China) with paired-end reads: 28 cycles for read 1, 90 cycles for read 2, and 10 cycles for i7 and i5 indexes. Single-cell RNA sequencing was carried out using the equipment of the Core Facility “Medical Genomics” (Tomsk National Research Medical Center, Tomsk, Russia).

### 2.3. Single-Cell Transcriptome Analysis

Raw sequencing reads were aligned to the human reference genome (GRCh38/hg38) with the addition of the HPV59 genome sequence (GeneBank accession LR862080) as a separate chromosome and processed using seeksoultools 1.2.0 (Beijing SeekGene BioSciences, Shenzhen, Guangdong, China). Results of seeksoultools processing were analyzed with Seurat 5.3.0 (Center for Genomics and Systems Biology, New York, NY, USA), R 4.4.0, Biocinductor 3.20.

Differential gene expression analysis was performed using either Seurat or DESeq2 1.36.0 (for aggregated SC data) as indicated in the figure and table legends with Benjamini–Hochberg multiple comparison correction by default. Stem-like cells as well as senescent-like cells were identified using the AUCell algorithm [22] with default parameters (AUCell thresholds 10% of expected cells) and custom gene signatures. Cell–cell interactions were constructed with CellChat V2 [23,24]. Cell cycle analysis was performed using the Seurat CellCycleScoring function with default parameters. Gene enrichment analysis of up- and downregulated genes was performed with Enrichr [25,26] that uses Benjamini–Hochberg multiple comparison correction by default. CNV events were analyzed using InferCNV [27,28]. Schemas were created with Cytoscape 3.10.3 [29].

### 2.4. Cytokine Concentrations in AdMer35 and AdMer43 Cell Media

Cells were seeded in 60 mm Petri dishes at a density of 5 × 10^5^ cells per dish and cultured until fully confluent, which was typically achieved within 72 h. The conditioned medium was then collected, centrifuged at 4000 rpm to remove cell debris, and stored at −70 °C until analysis. Cytokine and growth factor concentrations in the cell media were measured using the MILLIPLEX MAP Human Cytokine/Chemokine Panel (HCYTA-60K-PX38, Merck Millipore, Darmstadt, Germany) according to the manufacturer’s instructions. Briefly, 25 µL of assay buffer was added to each well of a 96-well plate, followed by 25 µL of standards, quality controls, or samples. Next, 25 µL of mixed magnetic beads precoated with capture antibodies specific for each analyte was added to each well. The plate was sealed, protected from light, and incubated overnight at 4 °C on a plate shaker. After incubation, the wells were washed twice with wash buffer, and detection antibody mixture was added to each well and incubated for 1 h at room temperature with constant shaking. Then, streptavidin–phycoerythrin (PE) conjugate was added to each well and incubated for 30 min under the same conditions. After two additional washes with wash buffer, sheath fluid was added to each well, and the beads were resuspended for 5 min before data acquisition using QuattroPlex (Moscow, Russia). All samples and standards were analyzed in duplicate, and the intra- and interassay coefficients of variation were <10% and <20%, respectively, as specified by the manufacturer.

## 3. Results and Discussion

Most cases of early-stage invasive cervical cancer can be cured with a combination of surgery, radiation, and chemotherapy. However, some tumors recur quickly and, in most cases, are fatal despite chemotherapy. With that, little is currently known about the biological mechanisms that might explain these differences in clinical behavior [30].

In this study, we performed a detailed analysis of single-cell transcriptome data from two cervical cancer cell lines derived from tumors before (Admer35) and after (AdMer43) chemoradiation therapy [17].

We use the term “chemoradioresistance” to describe the resistance phenotype that developed after combined chemoradiotherapy, consistent with the clinical treatment received by the patient from whom the AdMer43 cell line was derived. The terms “chemoresistance” and “radioresistance” are used only when referring specifically to effects related to chemotherapy or radiotherapy alone. Throughout this manuscript, unless explicitly stated otherwise, resistance refers to chemoradioresistance.

### 3.1. Cervical Cancer Cell Lines AdMer35 and AdMer43 Have Epithelial Cell Morphology

Both cell lines were derived from a 22-year-old patient diagnosed with stage IIbN1M0 (IIIB) squamous cell carcinoma of the cervix, exophytic form, vaginal–parametrial variant, and were obtained as described earlier [17]. Cell lines AdMer35 and AdMer43 grew attached to the flask; cells formed an irregular island pattern with a cobblestone morphology, which is characteristic of epithelial cells (Figure 1A). We showed that AdMer35 cells demonstrated more intensive growth of primary tumor nodes, which is associated with their high motility and migration activity found in vitro and can explain the ability to invade the surrounding tissues more effectively in vivo. By contrast, AdMer43 cells characterized by higher proliferative potential but a relatively low migration rate in vitro, in vivo form smaller, slowly growing tumor nodes as compared to AdMer35 [17]. At the same time, AdMer43 cells exhibited higher chemo- and radioresistance in vitro compared with AdMer35.

### 3.2. Single-Cell Transcriptomic Profiling of AdMer35 and AdMer43 Cells

AdMer35 and AdMer43 cell lines, as mentioned above, are cells obtained from a tumor after three cycles of polychemotherapy (AdMer35) and cells derived from a recurrent tumor after RT supplemented with chemotherapy (AdMer43) (see Table 1 for details). Being morphologically homogeneous (Figure 1A), these cells might differ in their molecular status. This is the main reason why we applied single-cell deep transcriptomic and subsequent bioinformatics analysis to identify and characterize key differences between cervical cancer cells of the same patient before and after chemoradiotherapy (Figure 1B).

The approach involves sample preparation and single-cell transcriptome sequencing on the Genolab M platform and alignment of experimental RNA sequences using SeekSoulTool with the combined human genome and HPV59 viral genome (hg38/HPV59); RNA mapping results were analyzed using the Seurat 5.3.0 software package [31]; CNV events were analyzed using InferCNV [27]. Stem-like cells as well as senescent-like cells were identified using the AUCell algorithm [22]. Cell cycle phase analysis was performed using Seurat’s CellCycleScoring function. Seurat’s FindMarkers and FindAllMarkers functions were used to analyze differentially expressed genes (DEGs) in different cell groups. To identify the biological functions, pathways, and processes in cells, gene enrichment analysis of DEGs in Enrichr was performed [25,26] (Figure 1B).

To perform an initial analysis of single-cell transcriptomic data for AdMer35 and AdMer43 cells, we compared the overall characteristics of the sequencing results (Table 1, Figure 1C). Although the total read counts for the AdMer35 and AdMer43 sequencing libraries differ by approximately 1.6-fold (Table 1), the initial results of the overall single-cell transcriptomic data suggest that both cell lines exhibit similar characteristics. The single-cell sequencing data obtained for AdMer35 and AdMer43 cells are characterized by similar distributions of the number of RNA molecules detected per cell and similar contributions of mitochondrial and viral HPV59 transcripts to the cellular transcriptomes (Figure 1C). Thus, the similarity in the overall characteristics of transcriptomic data from individual AdMer35 and AdMer43 cells allows for reliable subsequent comparisons between pre- and posttherapy transcriptional profiles.

### 3.3. Differentially Expressed Transcripts in AdMer43 Versus AdMer35 Cervical Cancer Cells

One of the most informative approaches to understanding the processes leading to cancer cell resistance to therapy is a detailed analysis of changes in cell transcriptomes after therapeutic intervention. To identify and characterize global changes in gene expression profile in obtained cell lines following chemoradiotherapy, we analyzed DEGs in AdMer43 vs. AdMer35 using “bulk” cellular transcriptomes. Differential gene expression analysis identified 1085 upregulated and 519 downregulated RNAs in AdMer43 vs. AdMer35 with the following parameters: |avg_log2FC| > 1.2, *p*_val < 1 × 10^−5^.

To construct a non-redundant gene network characterizing key transcriptional changes in AdMer43 cells compared to AdMer35 cells, we used gene enrichment analysis for up- and downregulated RNA sets using the Enrichr platform. The gene network included selected sequences, transcription factor (TF) -> gene/transcript -> biological characteristics, to represent the most diverse and non-redundant set of biological characteristics. To describe the potential TFs, we used gene enrichment analysis with the “ENCODE and ChEA Consensus TFs from ChIP-X” Enrichr library. With Enrichr analysis of AdMer43’s top 300 upregulated RNAs, it was determined that transcripts that are elevated in chemoradioresistance cells are controlled by a set of TFs shown in Figure 2A (yellow ovals). It should be mentioned that some TFs were activated because transcripts controlled by these TFs are upregulated, but the level of mRNA of the TFs themselves did not change significantly in the obtained datasets. For some other TFs, upregulation of both their mRNAs as well as their controlled mRNAs was observed. Also, there were TFs for which only upregulation of their mRNAs was found. Among activated TFs there are POU5F1, TCF3, GATA2, SUZ12, and EZH2. It was previously shown that activation of these TFs promotes tumor growth, proliferation, and invasion and is associated with a poor prognosis for patients [32]. For example, in previous investigations it was demonstrated that TCF3 promotes the growth, migration, and invasion of cervical cancer cells by activating the Wnt/β-catenin pathway, and high TCF3 levels in patients are correlated with more advanced clinical stages and reduced overall survival [32]. It was also shown that overexpression of transcription factor POU5F1 (also known as OCT4) in cervical cancer is linked to its development, progression, and potential resistance to treatment as well as to EMT [33].

We revealed that, with the activation of TF GATA2, an increase in the mRNA levels encoded by TFs belonging to the same zinc-finger TF family, GATA4 and GATA5, was observed (Figure 2A). As has been shown earlier, this is positively correlated with thyroid hormone receptor interactor 4 (TRIP4) expression, and high expression of both factors predicts poor prognosis in patients [34].

Along with the mentioned TFs, activation of two key polycomb repressive complex 2 (PRC2) components, SUZ12 and EZH2, was detected in AdMer43 cells (Figure 2A). The genes of the HOX family, which is under the control of PRC2, including HOXA10, -11, -13; HOXB4; HOXC4, -6, -8, -9, -10, -12 (Figure 2A), are transcription factors that are active in the development of embryonic stem cells [35]. HOX family members, namely HOXA10, -13; HOXC8 -9, -11; and others as previously shown, are dysregulated in HPV16-positive cervical cancer cells [36]. The deregulated expression of HOX cluster members in these cells is associated with EMT [36]. Moreover, transcripts of FOXF1, -A3, and -N4, which are upregulated in AdMer43 cells (Figure 2A), are associated with the development of metastases and EMT [37].

In AdMer43 cells, not only did activation of TF interferon regulatory factor 8 (IRF8) take place but an increased level of IRF8 mRNA itself was also detected. Since IRF8 modulates the expression of genes stimulated by interferons [38], it largely explains the increased expression in groups of genes associated with interferon alpha/gamma response, cytokine signaling, and inflammatory response (Figure 2A). We showed that in AdMer43 cells the mRNA levels of SRY-box TFs SOX2 and SOX5 are upregulated, but we did not observe an increase in activity of these TFs. SOX2 is known as a key regulator of cancer progression in a wide range of cancer types, while the role of SOX5 is currently poorly understood [39,40]. Thus, in AdMer43 cells we mainly observe an increase in the level of those mRNAs that are associated with pro-inflammatory pathways, tumor development, and metastasis formation.

Analysis of the top 300 upregulated transcripts in AdMer43 cells shows that this set is enriched with mRNAs encoding non-fibrillar collagens COL18A1, COL26A, and COL23A1 and mRNAs encoding proteins involved in inflammatory response, cytokine-mediated signaling, various interferon responses, and signaling, as well as extracellular matrix structure organization, etc. (Figure 2A, magenta and white rectangles).

Analysis of the list of transcripts from AdMer43 cells in the Enrichr “TargetScan microRNA 2017” library (adjusted *p*-value < 0.05) showed that the expression of some genes among the top 300 is controlled by two microRNAs: hsa-miR-4508 and hsa-miR-3180-3p (Figure 2A). These microRNAs share both common and unique mRNA targets. Their common mRNA targets are NPTX1, -2. Neuronal pentraxin 1 (NPTX1) is known as a cancer-associated protein overexpressed in human and murine cells during metastasis [41]. NPTX2 promotes colorectal cancer growth and liver metastasis [42] and renal cell carcinoma proliferation, migration, and invasion [43].

Hsa-miR-3180-3p itself regulates HOXC6, -10, as well as COL18A1 mRNA translation (Figure 2A). Expression of miR-3180-3p is generally reduced in non-small cell lung cancer tissues, while increased levels of miR-3180-3p suppress cancer cell proliferation, migration, and invasion [44]. Since in our case we observe an increased level of mRNA transcripts regulated by hsa-miR-3180-3p, it can be assumed it has pro-oncogenic functionality in the transcriptomic context of chemoradioresistant AdMer43 cells. Among targets of miR-4508 is BCL11B whose overexpression leads to anti-apoptotic signal activation and induces resistance to chemotherapy, affecting the postoperative prognosis [45]. This miR-4508 is associated with cancer progression, playing a role in cell proliferation, invasion, metastasis, and the formation of premetastatic niches [46,47].

It should also be mentioned that AdMer43 cell transcripts with increased expression are enriched in the products of genes encoded by chromosomes 12 and 19 (Figure 2A).

Similar analysis was performed for the top 300 transcripts that are downregulated in AdMer43 cells. We found that these transcripts are controlled by a set of TFs (Figure 2B, yellow ovals), including STAT3, NANOG, SOX2, TP53, and TP63. Most of them are known to be involved in carcinogenesis and cervical cancer cell proliferation [48]. Our data revealed a combined downregulation of activities of stem-cell-related TFs STAT3, NANOG, and SOX2 (Figure 2B), which means that most, but not all, cells in the population share reduced stem cell characteristics compared to AdMer35 cells. It is important to note that in AdMer43 cells we also detected downregulation of genes involved in EMT, such as TPM1, VCAN, COL1A2, MMP2, and others (Figure 2B).

Inactivation of TP53 is a well-known hallmark of cervical carcinogenesis via HPV E6/E7 activity. Viral oncogenes E6 and E7 are the key mediators of oncogenic transformation of cervical cells through disruption of the TP53 and RB pathways [30]. Obtained data show that activity not only of TP53 but also of TP63 is downregulated in cervical cancer cells resistant to chemoradiotherapy (Figure 2B).

Similar to transcript sets that were increased in AdMer43 cells, transcript sets with decreased expression were enriched with mRNAs encoding membrane proteins, extracellular matrix, and cell adhesion proteins (Figure 2A,B). In this context, mRNAs encoding collagens COL18A1, -23A1, and -26A1 are upregulated (Figure 2A), while levels of COL1A2, -11A2, -25A1, and -14A1 mRNAs are decreased (Figure 2B). Collagens contribute to the extracellular matrix (ECM) stiffness and remodeling, both being crucial for cancer cell attachment, proliferation, migration, and invasion, all of which malignant tumors exploit for growth and dissemination [49]. The collagens whose mRNAs were upregulated in AdMer43 cells were non-fibrillar collagen types—COL18A1, -23A1, and -26A1—while the downregulated mRNAs (COL1A2, -11A2, and -14A1) are those encoding fibrillar collagen, minor fibrillar collagen, and collagen associated with fibrillar protein, and only COL25A1 is non-fibrillar collagen (classified according to [50]). Thus, resistance to chemoradiotherapy of AdMer43 cells may be partly explained by a switch in the expression of fibrillar collagens to non-fibrillar ones.

COL18A1 is a precursor of endostatin, a potent antiangiogenic protein capable of inhibiting angiogenesis and tumor growth. Endostatin is currently considered a component of a complex that includes a balance of pro- and anti-angiogenic/metastatic factors [51]. COL23A1 is described as a pro-oncogenic factor in clear cell renal cell carcinoma [52]. Our study shows that the mRNA of fibrillar COL1A2 is downregulated in AdMer43 cells (Figure 2B). Since it is known that COL1A2 has a close interaction with COL1A1 which has been proposed as a key factor in cervical cancer cell radioresistance and to play an important role in suppressing apoptosis via caspase-3/PI3K/AKT pathways [49], we can suggest COL1A2 is another factor also suppressing CC resistance to chemoradiotherapy. Thus, our data support the hypothesis that tumor chemoradioresistance is determined, to some extent, by individual factors, such as COL1A1 or COL18A1, but also by a delicate balance of particular expression of membrane proteins, extracellular matrix proteins, and proteins of cell adhesion.

A broad set of mRNAs downregulated in AdMer43 are enriched for genes controlled by the hsa-mir-190b microRNA, including TCF4, AKT3, MMP2, and others shown in Figure 2B. MicroRNA-190b (miR-190b) is aberrantly expressed in multiple types of cancers. Expression of this miRNA was significantly increased in colorectal and other cancers, suggesting that miR-190b functions as an oncogene [53]. Thus, downregulation of a number of mRNA targets of hsa-mir-190b (Figure 2B) indirectly confirms the pro-oncogene properties of this microRNA in chemoradioresistance development of cervical cancer cells.

Also, in AdMer35 cells the decreased expression of RNA transcripts related to chromosome 4, as well as to chromosome 9 loci and chromosome 18 (Figure 2B), was observed, so it can be associated with copy number variation (CNV) loss events in these chromosomes.

Taken together, our DEG analysis indicates that AdMer43 cell resistance to chemoradiotherapy is characterized by a general switch in the expression of fibrillar collagens to non-fibrillar ones, increased mRNA levels of embryonic stem cell developmental TFs of the HOX and SOX families, and activation of innate immune/inflammatory response genes. AdMer43 cells also exhibit decreased activity of stem-cell-associated TFs: STAT3, NANOG, and SOX2, decreased expression of genes involved in epithelial–mesenchymal transition, and decreased expression of genes involved in the EGF, FGF, Wnt, and JAK-STAT signaling pathways.

### 3.4. AdMer35 and AdMer43 Cells Differ in CNV Events in Multiple Chromosomes

To identify the major cell subpopulations in AdMer35 and AdMer43 lines, we compared their transcriptomes using non-linear dimensionality reduction with uniform manifold approximation and projection (UMAP) (Figure 3A). In the UMAP plot, constructed with the DimPlot Seurat function, transcriptomes of AdMer35 and AdMer43 cells were found to be clearly separated into two large groups representing the majority of cells in these lines. With that, mixed subpopulations of AdMer35/43 cells were identified that had similar characteristics of transcriptomes for both AdMer35 and AdMer43 in terms of UMAP coordinates (Figure 3A).

Using the InferCNV [27], which implements a hidden Markov model to find CNV events in single-cell transcriptomic data, we identified a number of potential genomic aberrations that distinguish AdMer35 and AdMer43 cells. Data for non-malignant cervical cells were used as a control [16]. From the data displayed in Figure 4, it is evident that most of the CNV clusters (>95%) of AdMer35 and AdMer43 cells are clearly distinct from each other and form two separate branches on the hierarchical tree (Figure 4A).

Cells of both studied lines are heterogeneous in CNV events and are divided into dozens of subpopulations with specific patterns of these events. With that, it is clear that both AdMer35 and AdMer43 cells share a range of CNV events. So, gains in chromosome arms 8q, 9q, and 19p, as well as the losses in loci of 8p and 11q, are common CNV events as compared to control non-malignant cells (Figure 4A). In this patient-derived model, for the majority (>80%) of AdMer35 cells, specific gains in chromosome loci 2q, 13q, and 18p were shown. As for AdMer43 cells, specific gains in 5q and 12p and loss events in 3p, 4p, 4q, and 18q were observed (Figure 4A). These CNV patterns likely reflect patient-specific clonal evolution rather than recurrent alterations across cervical cancer.

The CNV event detection data by InferCNV (Figure 4A) are in good agreement with the DEG analysis by Enrichr, confirm changes in loci of chromosomes 4, 9, 12, and 18 (Figure 2), and can be explained by CNV events as chromosomal aberrations in AdMer35 and AdMer43 cells.

It is important to note that some InferCNV-identified subpopulations of AdMer35 and AdMer43 cells share a common CNV (Figure 4A, horizontal black lines). The mixed AdMer35/43 (Figure 4A,B, “AdMer35/43” cells) subpopulation is characterized by cells with lower inferred CNV burden compared to the control and contains cells from both cultures, so these cells can be considered as possible progenitor-like cells (PLCs).

If we compare the data in Figure 3A,B, the highest concentration of PLCs is located in the region of overlapping UMAP coordinates of mixed AdMer35/43 cell populations.

Overall, these data show that chemoradiotherapy resulted in significant alteration of chromosomal structure, likely resulting from both chromosomal aberrations and selection of new cervical cancer cell subclones after the therapy. Importantly, although the CNV patterns we found are patient-specific, our analysis can be used alongside many other single-cell transcriptomic approaches to study therapy-driven clonal evolution within an individual malignancy. Detailed analysis of cell subpopulations allows identification of cervical cancer cells with lower inferred CNV burden, which are considered as possible PLCs, with a CNV pattern similar to but not equal to the non-malignant karyotype.

### 3.5. Senescent-like Cells (SnCs) in AdMer43 and AdMer35 Lines

Increased oxidative stress associated with DNA damage, oncogene activation, and therapeutic interventions such as radiation is a well-known inducer of cellular senescence [54]. However, it is important to emphasize that the observations described below are based on a single-patient, paired isogenic model and therefore reflect case-specific cellular adaptations to chemoradiotherapy rather than universally applicable mechanisms of chemoradiotherapy resistance in cervical cancer.

Senescence is one way in which tumor cells evade the direct cytotoxic effects of radiation and chemotherapy, allowing them to remain dormant for extended periods of time and potentially restore their ability to self-renew and promote tumor recurrence and metastasis [10]. In order to identify and describe senescent-like cells (SnCs) within AdMer35 and AdMer43 cell populations, we used the AUCell algorithm with custom RNA markers chosen using literature data and the Enrichr “WikiPathways 2024 Human” library “Senescence Associated Secretory Phenotype SASP WP3391” (Appendix A) [55,56,57,58,59,60,61,62,63,64].

Performed analysis of marker expression within obtained datasets showed that SnCs can be found both in AdMer43 and AdMer35 cultures with similar relative contributions of about ~10.2 and 9.8%, respectively (Figure 3C). It should be noted that the enriched PLC subpopulation was more than double in SnCs (Table 2) and may represent a reservoir of cellular plasticity that survives therapy and could contribute to tumor recurrence.

When analyzing the cell cycle phases with Seurat CellCycleScoring, it was found that SnCs are characterized by an arrest in the G1 phase and reduced relative content of cells in the G2M division phase of the cell cycle compared to non-SnCs; that is one of the hallmarks of senescence [54] (Table 3). The relative contribution of SnCs in the G2M phase is more than three times lower than in the sum of G1 and S phases. These data confirm that both cervical cancer cell lines retain a set of silent, dormant cells with reduced division potential and retain a malignant karyotype and phenotype.

Thus, mixed populations of AdMer35 and AdMer43 cells with similar transcriptomic patterns are represented simultaneously by cells with a large contribution of PLCs and SnCs (Figure 3B,C, Table 2).

To understand how similar SnCs are in AdMer35 and AdMer43, we determined differentially expressed transcripts (top 300 up- and top 300 downregulated, with |avg_log2FC| > 1.2, *p*_val < 1 × 10^−5^) in SnCs vs. non-SnCs in the AdMer35 and AdMer43 datasets separately (Figure 5). We found 50 common upregulated and 48 common downregulated transcripts.

WikiPathways 2024 Human analysis showed that all common upregulated transcripts are associated with the “Senescence and autophagy in cancer” WikiPathway. Among the most important factors, we can highlight the activation of transcription factor ZC3H11A which is a critical component of the TREX complex involved in both pol II transcription and mRNA export from the nucleus to the cytoplasm [65]. It plays an important role in the regulation of NF-κB signaling [66]. Also, we observed that a senescence-like state in cervical cancer cell lines is commonly associated with the activation of transcription factors ZC3H11A and RELA (Figure 5A).

Common downregulated transcripts are characterized by “DNA damage response” in WikiPathways (Figure 5A). Our data showed that all AdMer43 and AdMer35 SnCs exhibited reduced Forkhead box M1 (FOXM1) TF activity. FOXM1, a member of the Forkhead superfamily of transcription factors, drives G2M phase transition and has been shown to be a key hierarchical-level factor in chromatin remodeling required to prevent aging [67]. Both SnC populations are characterized by common downregulation of genes involved in p53 signaling pathways and cyclin mRNAs regulating the delay the onset of cellular aging, as well as lowered expression of DNA damage response genes (Figure 5A).

The most interesting question is the transcripts, TFs, and process that allow us to distinguish SnCs before and after chemoradiotherapy. In chemoradioresistant AdMer43 SnCs, a shift from genes controlled by the TFs TP53 and NFE2L2 to genes controlled by the TFs NELFE, REST, and RELA was observed. It should be noted that TF RELA plays a crucial role in cellular senescence, primarily by regulating the senescence-associated secretory phenotype (SASP) and maintaining genomic stability [68]. This TF is also activated in the common population of SnCs. Comparison of SnC populations from AdMer35 and AdMer43 shows that a transition from the inflammatory response and estrogen response to Wnt-beta and IL-6/JAK/STAT3 signaling takes place. SnCs from AdMer43 are characterized by changes in the levels of transcripts encoding mitochondrial components. Specifically, there is an increase in mRNAs whose products are involved in apoptotic mitochondrial changes, while the contribution of mRNAs encoding mitochondrial membrane components, oxidative phosphorylation, and the electron transport chain is reduced. This indicates the processes of mitochondrial dysfunction characteristic of SnCs [54] (Figure 5A).

Analysis of intercellular communication is one of the most promising approaches to the detailed description of the diversity of processes in organs, tissues, and cells. To describe cross-talk patterns between specific ligands and cellular receptors, we used CellChat V2, a tool capable of quantitatively inferring and analyzing intercellular communication networks from single-cell RNA-sequencing data. It should be noted that CellChat-based ligand–receptor interaction analysis is predictive in nature and relies on gene expression patterns with carefully curated interaction databases. The current version of CellChat V2 with CellChatDB v2 contains approximately 3300 validated molecular interactions, which include protein and non-protein interactions like synaptic and metabolic signaling [23,24].

Using CellChat, we analyze intercellular communication in single-cell datasets of SnCs in the AdMer43 and AdMer35 cultures. Both cell lines were found to be characterized by extensive patterns of cross-talk between SnCs and non-SnCs (as well as SnC–SnC and non-SnC–non-SnC) at the level of MDK as a ligand and SDC2, -4, ITGA4, -6,-B1, and LRP1 as receptors (Figure 5B, Appendix A).

The midkine (MDK) gene encodes secreted growth factors that bind heparin. The encoded protein promotes cell growth, migration, and angiogenesis, particularly during tumorigenesis [69]. MDK receptors include integrins, Notch2, syndecanes SDC2/4, glipican-2, PG-M/versican, and neuroglycan C, anaplastic lymphoma kinase, the low-density lipoprotein receptors, and receptor-type tyrosine protein phosphatase zeta (Appendix A). MDK binds to syndecans, which are membrane-associated proteoglycans, creating a complex that initiates signaling events via pathways like MAPK/ERK, PI3K/AKT, and STAT3, leading to uncontrolled cell growth and tumor progression and regulating drug resistance. MDK may produce either a drug-resistant or drug-sensitive cancer cell phenotype in different conditions [69].

The data in Figure 5B show that cell–cell communication of AdMer43 SnC differs significantly from that of AdMer35. AdMer35 SnCs predominantly exhibit cross-talk involving PPIA/BSG (CypA), PLAU/PLAUR (PLAU), and NAMPT/ITGA5+ITGB1 (VISFATIN). SnCs in chemoradioresistant AdMer43 culture switch to cell–cell interactions involving EREG/EGFR (EGF), SEMA3C/NRP1, PLXNA1, -2, -3, PLXND1 (SEMA3C), BMP2/BMPR1A, -2 (BMP), and WNT11/FZD6, -10 (WNT) (Figure 5B, Appendix A) [70,71,72,73,74,75,76,77,78,79,80,81].

As mentioned above, analysis reveals a coordinated pro-tumorigenic network in cervical cancer cell–cell communications mediated by four key ligands (Figure 5B). It is known that EREG, an EGFR ligand, promotes proliferation and modulates the tumor microenvironment through interactions with immune cells, fibroblasts, and endothelial cells [82,83]. SEMA3C drives EMT, TGFβ signaling, angiogenesis, and ECM interactions, all associated with poor prognosis [84]. BMP2 activates SMAD signaling and is linked to cancer stem cell genesis, EMT, and chemotherapy resistance in gynecological cancers [85]. WNT11 promotes tumor proliferation and migration/invasion through Wnt/JNK pathway activation, with its elevated expression correlating with poor outcome [86,87]. The simultaneous upregulation of these ligands creates a synergistic effect, where their complementary pathways collectively drive tumor aggressiveness and treatment resistance.

Taken together, our data show that the dormant-like cell subpopulation of the chemoradioresistant cell line AdMer43 exhibits reduced levels of mRNA encoding components of mitochondrial oxidative phosphorylation. Activation of Wnt signaling and switch of inter/intracellular communication to EREG, SEMAC, and BMP2 might be a reason for the developed chemo- and radioresistance.

### 3.6. Cervical Cancer Stem-like Cells in AdMer43 and AdMer35 Lines

One of the key issues for understanding tumor cell proliferation, tumor growth, invasion, and metastasis is the identification and characterization of cancer stem cells in cell populations. To identify cancer stem-like cells (SLCs) in the AdMer43 and AdMer35 cultures, we used the AUCell approach [22] with a custom set of molecular markers of cancer stem cells selected from literature data (Appendix A) [88,89,90,91,92,93,94,95,96,97,98,99]. Analysis of marker expression allowed us to identify SLCs characterized by elevated levels of these transcripts among AdMer43 and AdMer35 cells (Figure 3D).

Performed AUCell analysis showed that SLCs in UMAP coordinates are distributed unevenly within AdMer35 and AdMer43 cell populations, with a higher abundance of SLCs in the AdMer35 cell line (Table 4, Figure 3D).

When we compared the contributions of SLCs with the contributions of SnCs, we found that the SLC population is not enriched in SnCs, and, conversely, SnCs exhibit a low SLC content, approximately 10–13% (Appendix A). Thus, this percentage indicates that the characterization and the assignment of cells as SLC or SnC occur independently and these are different cell populations in AdMer35 and AdMer43 cell cultures.

To characterize the similarities and differences in SLC characteristics between AdMer35 and AdMer43 cultures, we identified differentially expressed transcripts (top 300 upregulated and top 300 downregulated |avg_log2FC| > 1.6, *p*_val < 1 × 10^−5^) in SLCs vs. non-SLCs in the AdMer35 and AdMer43 datasets independently (Figure 6A). We found that 57 upregulated transcripts common for SLCs from both cell cultures are enriched for genes classified by MSigDB as “Epithelial–Mesenchymal Transition” and by Panther as “Cadherin Signaling Pathway.” Transcripts whose expression was generally downregulated in SLCs show enrichment for the MSigDB terms “Angiogenesis” and “p53 Pathway” as well as for the GO term “Regulation of DNA Damage Response” (Figure 6A).

Thus, both AdMer35 and AdMer43 SLCs exhibit elevated levels of EMT-related mRNAs, reduced expression of p53-regulated DNA repair genes, and reduced levels of DNA-damage-responsive transcripts.

Among the most important factors, we can highlight the elevated levels of transcripts controlled by TFs TP63 and AR in the common population of SLCs (Figure 6A). With that, the data show that the chemoradioresistant SLCs of AdMer43 have at least partially switched from transcriptional regulation by the TFs FOSL2, TCF3, and NANOG to genes controlled by SMAD4, SOX2, and POU5F1.

A common feature of SLCs of the two lines was reduced mRNA levels for the “p53 Pathway,” “Regulation of DNA Damage Response,” and “Signal Transduction by P53 Class Mediator Processes,” and given that the TF TP53 is also reduced in SLCs of AdMer43, then, apparently, an even deeper reduction of the DNA repair process occurs in these cells (Figure 6A).

CellChat V2 analysis of cell–cell communication of SLCs from AdMer43 and AdMer35 lines shows that cells of both cultures are characterized by extensive patterns of cross-talk at the level of VISFATIN, GAS, EGF, and SEMA3, BMP pathways (Figure 6B, Appendix A). Chemoradioresistant AdMer43 SLCs exhibit a wide range of cellular interactions. This includes the MK pathway, which can induce both drug-resistant and drug-sensitive phenotypes of cancer cells under various conditions [70]; the TGFb pathway, which can both promote and slow down the progression of cervical cancer [100]; EREG/EGFR (EGF pathway); wcWNT, PLAU, GRN, and others (Figure 6B, Appendix A).

Comparing the cellular communications specific to SLCs (Figure 6B) and those for SnCs (Figure 5B) in AdMer43 cells, one can see that these subpopulations do not share common specific signaling pathways except ncWNT, which promotes tumor cell proliferation and migration in cervical cancer (Appendix A). Thus, in this patient-derived model, chemoradioresistance of AdMer43 cells appears to be associated with at least two distinct levels of cellular organization: intercellular communication involving stem-like cells and senescent-like dormant cell populations.

Taken together, SLC demonstrates distinct molecular profiles between the AdMer35 and AdMer43 lines. While both SLC populations share elevated EMT signaling and suppressed DNA damage response pathways, chemoradioresistant AdMer43 SLCs show specific transcriptional reprogramming toward SMAD4/SOX2/POU5F1 regulation and a switch from estrogen response to KRAS signaling with a shift towards pro-tumorigenic communication through MK, TGFβ, EGF, and WNT pathways. These SLC-specific alterations represent a key mechanism of therapy resistance complementary to senescence-associated pathways.

### 3.7. Generalized Data on the Classification of AdMer35 and AdMer43 Cultures

To account for the most important cellular characteristics and avoid creating new redundant clusters, we minimized the number of Seurat clusters to integrate a set of different cell classifications, such as SLCs, SnCs, cell phases, and others. As a result of the optimization, we obtained six clusters (Figure 7A, Table 5). Such classification (numbered 0 to 5) allows us to clearly separate AdMer35 and AdMer43 cell lines based on their RNA expression patterns. Clusters 0 and 3 were found to be specific for AdMer35, while clusters 1 and 2 were for AdMer43. Clusters 4 and 5 included cells from both cell lines (Figure 7A,B; see also Figure 3A). We can see that RNA expression patterns found for clusters 0 and 3 and 1 and 2 for AdMer35 and for AdMer43, respectively, are somewhat different but share some common features (Figure 7B, yellow spots on the heat map). This indicates that cells in cluster pairs 0 and 3 and 1 and 2 have common upregulated RNAs within each pair. Common characteristic transcripts are also found in clusters 2 and 3, which obviously belong to different cell lines (compare Figure 7A,B, yellow spots on the heat map). This means that the cells from different lineages apparently share common fundamental biological processes.

In order to describe the diversity of individual and general characteristics of cells in Seurat clusters, we create a table summing the main characteristics of each cluster (Table 5). As depicted in Table 5, clusters 0 and 3 are represented by AdMer35, while clusters 1 and 2 are represented by AdMer43 cells (>99%). Clusters 4 and 5 contain cells of both lines (Table 5, “Cell Lines”). Clusters 4 and 5 of the “mixed cells” differ significantly in their relative viral transcript abundance, both from each other and from other clusters. Cluster 4 cells contain ~0.1% viral RNA transcripts, while cluster 5 cells contain only 0.001%, which differs significantly from the average for the other clusters (~0.07%) (Table 5, “HPV59 Viral Transcripts”).

One of the most fundamental and critical characteristics of cell lines is the cell distribution according to the phases of the cell cycle. The cell cycle is not just a biological process but a central quality control metric. It determines a cell line’s growth capacity, genetic integrity, and functional utility, making its characterization indispensable for ensuring reproducible, meaningful, and translatable scientific outcomes. To classify the cells of the AdMer35 and AdMer43 lines by cell cycle phases, we used the Seurat CellCycleScoring approach with S and G2M cycle phase markers (Appendix A). It was found that both cell lines exhibited quite similar cell cycle phase distribution (Appendix A). However, analysis of cell distribution between phases of the cell cycle performed for clusters 0–5 shows that cells in clusters 2 and 3 are predominantly in the G2M phase (77% and 87%, respectively), while cells in clusters 0 and 1 are in the G1 and S phases. Cells in clusters 4 and 5 were evenly distributed between G2M, G1, and S phases (Figure 7C, Table 5, “Cell Cycle”).

It is important to highlight that cluster 5, represented by cells of both cervical cancer lines, is significantly enriched in cells whose karyotype is close to normal non-cancerous cells and which we designated as PLCs (~63% in cluster 5 vs. ~1% in other clusters) (Table 5, “Progenitor-Like Cells (PLCs)”). At the same time, the cells of cluster 5 are significantly depleted in cancer stem cells (0.72% vs. ~10% average for the other clusters) (Table 5, “Cervical Cancer Stem-Like Cells (SLCs)”). Cluster 5 is also enriched with SnCs (~27% vs. average ~10%) (Table 5). It is worth mentioning that a high abundance of SnCs was detected in cluster 4 also (~16%).

To summarize, clusters 0–3, containing the majority of cells in their lineages, are divided into pairs 0 and 3 (AdMer35) and 1 and 2 (AdMer43), mainly by cell cycle phases and corresponding processes: GO—Mitotic Sister Chromatid Segregation, Mitotic Spindle; MSigDB—E2F Targets, G2M Checkpoint, and others for actively dividing cells in the G2M phase for clusters 2 and 3 (Appendix A).

Differentially expressed transcripts of chemoradioresistant AdMer43 cells in G1/S phases are enriched with genes from the following groups: Interferon Alpha/Gamma Response, Inflammatory Response, KRAS Signaling Up, Response to Cytokine, Collagen Containing Extracellular Matrix, and Extracellular Matrix Organization (Appendix A, cluster 1).

At the same time, in cluster 0, consisting of AdMer35 cells in the G1/S phases, the expression of genes of the following groups is increased (and in AdMer43, correspondingly decreased): Epithelial–Mesenchymal Transition, Hedgehog Signaling, UV Response Dn, Chemorepellent Activity, Wnt Signaling, and Cadherin Signaling (Appendix A, cluster 0).

The most interesting cluster 5 is composed of cells from both cell lines enriched with PLCs, with low SLC content and high SnC contributions. Cells of this cluster are characterized by increased levels of mRNAs from the following groups: p53 Pathway; TNF-alpha Signaling via NF-κB; DNA Repair and Senescence Associated Secretory Phenotype (Appendix A, cluster 5). The opposite situation is observed for cluster 4, which also consists of cells from both lines. This cluster is enriched with SLCs, with low PLC content. Cells of this cluster are characterized by increased levels of mRNAs from the p53 pathway and KRAS signaling.

Thus, obtained data on the characteristics of AdMer35 and AdMer43 cells belonging to different Seurat clusters let us clearly identify differences in key biological processes occurring in chemosensitive and chemoradioresistant cervical cancer cell cultures AdMer35 and AdMer43.

### 3.8. Validation of Single-Cell Sequencing Data Using an Independent Cytokine/Chemokine Assay in AdMer35 and AdMer43 Cell-Condensed Media

To validate the adequacy of the single-cell transcriptome data, we analyzed the concentration of a panel of secreted cytokines and chemokines (see the Section 2 for details) in the culture media of AdMer35 and AdMer43 cells, followed by a comparison of RNA expression data and protein concentrations (Figure 8). We found that, in the culture media of AdMer35 and AdMer43 cells, the levels of a number of cytokines and chemokines are significantly increased (Figure 8, yellow and blue dots, Y axis). Among the cytokines measured, dramatically increased levels of CCL5 (RANTES) and IL6 are noteworthy. It has been shown that a high level of CCL5 is correlated with the progression of cervical cancer. CCL5 is regarded as a possible prognostic marker and therapeutic target since it plays a significant role in mediating inflammation and immunological response [101]. Moreover, it was shown that the simultaneous expression of IL-6 and RANTES, the levels of which were found to be increased in AdMer43 cell media, produces a more aggressive phenotype in the case of breast cancer cells [102]. Performed correlation analysis (Figure 8) shows very good coincidence of transcriptomic data and protein concentration. The Pearson correlation coefficient for cytokine expression in AdMer35 cells was R^2^ = 0.79, while for AdMer43 cells, R^2^ = 0.94, generally confirming the correlation between RNA-sequencing data and data of protein expression related to inflammation, immune response, and cancer progression.

## 4. Conclusions

Transcriptomic profiling has been used in deconvoluting the molecular pathogenesis of cervical cancer. Bulk RNA-sequencing studies have identified key pathways involved in HPV-induced transformation and progression [13]. More recently, single-cell RNA sequencing has provided unprecedented resolution, revealing the cellular heterogeneity of tumors, dynamic remodeling of the immune microenvironment, and critical stroma–tumor interactions [15,16]. However, the precise transcriptomic alterations that confer resistance to chemoradiotherapy at the single-cell level remain an area of active investigation.

In our study, single-cell transcriptome analysis of the unique paired cell lines before (Admer35) and after (AdMer43) chemoradiation therapy provides a multifaceted understanding of the molecular mechanism that determines the development of therapy resistance in cervical cancer. Our findings suggest that, in this model, resistance does not represent a single uniform state but rather a complex and adaptive phenotype accompanied by coordinated alterations across multiple cellular processes. Key drivers identified in the resistant AdMer43 cells include a significant transcriptional rewiring involving the activation of embryonic and stemness-associated transcription factors (e.g., HOX genes, POU5F1, SOX2), a profound shift in the extracellular matrix composition from fibrillar to non-fibrillar collagens, and a robust activation of innate immune and inflammatory response pathways. Concurrently, these cells exhibit a downregulation of critical tumor-suppressive pathways, including p53 signaling and DNA damage response mechanisms, and a reduction in epithelial–mesenchymal transition characteristics, suggesting a transition to a different, more resilient survival strategy.

The key aspect of our work is the detailed characterization of cellular heterogeneity and dynamic changes within subpopulations induced by therapeutic intervention. The analysis uncovered significant genomic evolution, evidenced by distinct copy number variation patterns in AdMer43, indicating clonal selection posttherapy. Furthermore, we identified and characterized two critical cell states: cancer stem-like cells (SLCs) and therapy-induced senescent cells (SnCs). While the overall proportion of SLCs was lower in AdMer43, the resistant line exhibited the processes of mitochondrial dysfunction characteristic of SnCs. Most notably, both SLCs and SnCs in AdMer43 displayed a switch in their intercellular communication networks, increasingly relying on pro-tumorigenic signaling pathways such as EREG/EGFR, SEMA3, BMP, and WNT. The discovery of a progenitor-like cell population with a near-normal karyotype, enriched in a mixed cluster with high senescence, suggests a dormant reservoir of cells that may survive therapy and serve as a foundation for tumor regeneration and recurrence.

In conclusion, this study integrates single-cell transcriptomic analyses to propose a conceptual framework describing cellular heterogeneity and inferred communication programs associated with chemoradioresistance in a patient-derived cervical cancer model. The identified transcriptional programs and predicted signaling pathways, including Wnt, TGFβ, BMP, and EREG/SEMA3C-related interactions, highlight candidate mechanisms that warrant further investigation.

## Figures and Tables

**Figure 1 cells-15-00115-f001:**
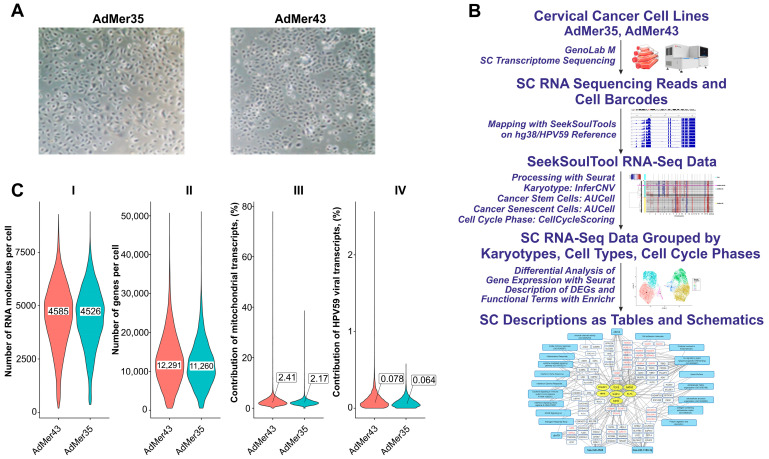
(**A**) Representative images of cervical cancer cell morphology: AdMer35 and AdMer43. (**B**) Experimental pipeline from obtaining cervical cancer cell lines to single-cell RNA sequencing, cell-type annotation by RNA patterns, and differential gene expression analysis focusing on cellular response to chemoradiotherapy. (**C**) Violin plots show the distribution of the number of genes per cell (I), the total number of RNA molecules detected per cell (II), the contribution of mitochondrial transcripts (III), and contribution of HPV59 viral transcripts to the cellular transcriptome in both single-cell sequencing arrays (IV). The median values of the distributions shown in insets (I,II) and callouts (III,IV).

**Figure 2 cells-15-00115-f002:**
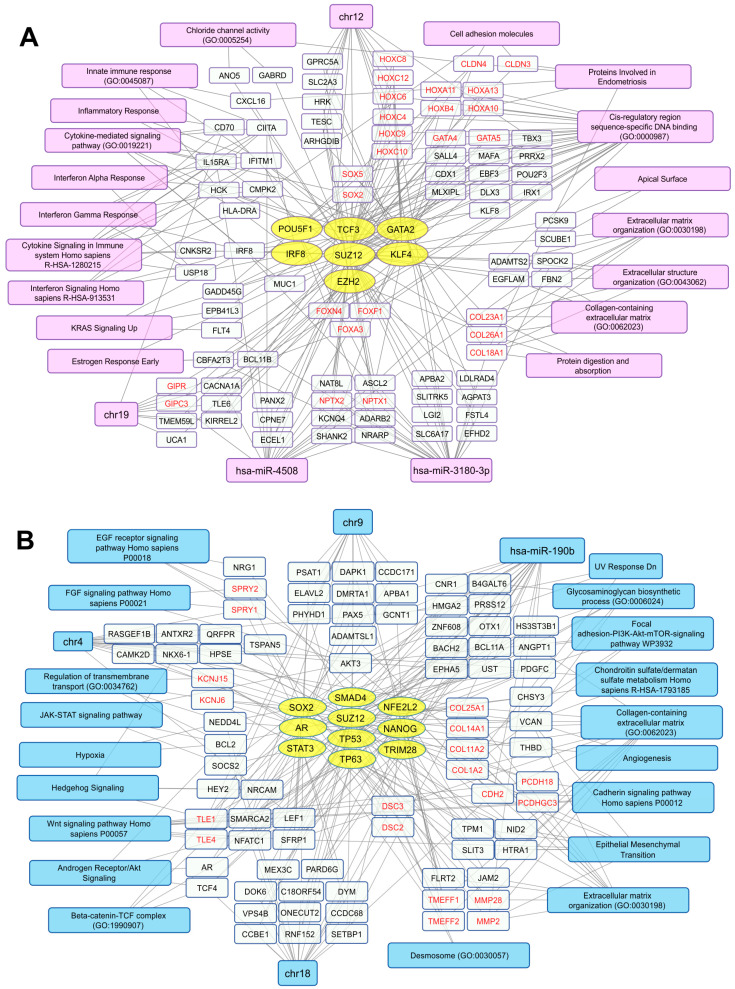
Transcriptional networks up- (**A**) and downregulated (**B**) in AdMer43 vs. AdMer35 cells. The network illustrates the relationships between transcription factors (yellow ovals), transcripts (white rectangles), and associated cellular processes or chromosome location (magenta or light blue). The analysis based on transcripts showing elevated (**A**) or reduced (**B**) level in AdMer43 vs. AdMer35 cells with Seurat SCTransform, FindMarkers functions (top 300 DEGs with |avg_log2FC| > 1.2, *p*_val < 1×10^−5^). Transcripts with similar gene functions grouped and highlighted in red text.

**Figure 3 cells-15-00115-f003:**
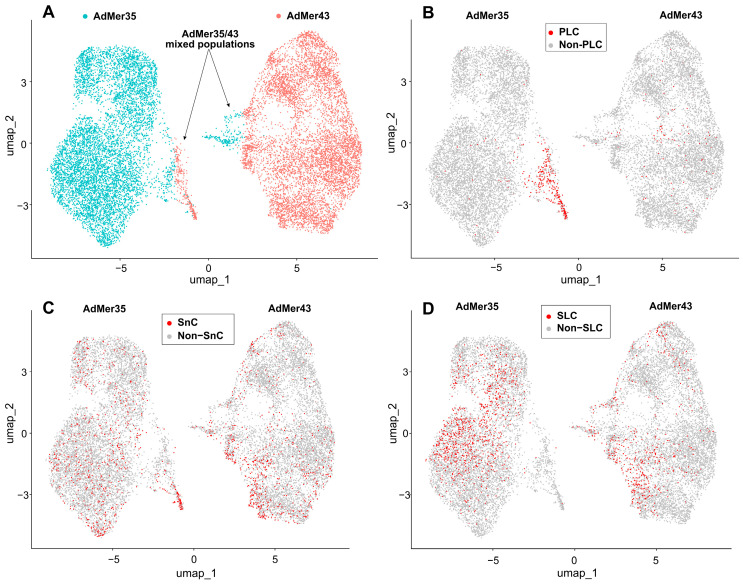
(**A**) Relationships between single-cell transcriptomes of AdMer35 and AdMer43 cells represented as non-linear dimensionality reduction UMAP plot. AdMer35 and AdMer43 highlighted with strong cyan and red dots, respectively. AdMer35 cells closely related to AdMer43 and vice versa designated as mixed AdMer35/43 populations and indicated by arrows. (**B**) Progenitor-like cells (PLC) from AdMer35/43 mixed subpopulation that share the CNV pattern close to the non-malignant one, (**C**) Senescent-like cells (SnC) and (**D**) Stem-like cells (SLC) are shown in the UMAP plots as red dots. All UMAP graphs plotted using Seurat’s DimPlot function.

**Figure 4 cells-15-00115-f004:**
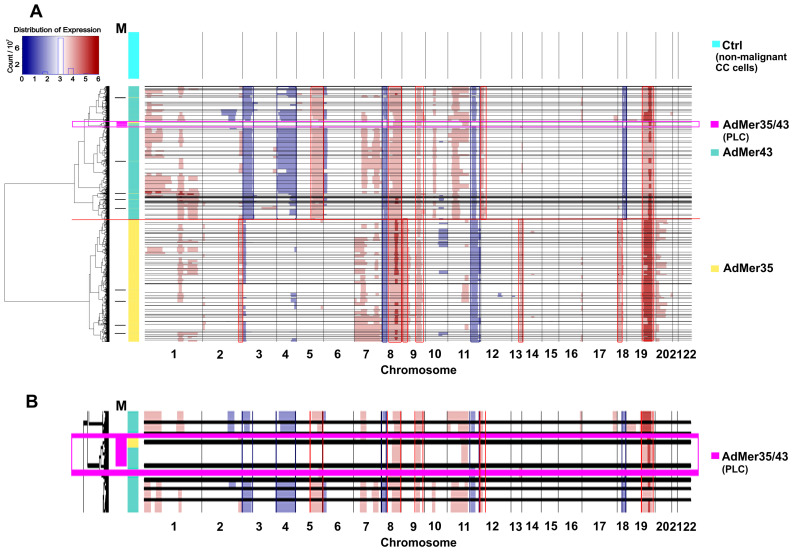
CNV events detected in AdMer35 and AdMer43 cells by InferCNV [27]. (**A**) Transcriptome data from non-malignant cervical cells were used as a control (Ctrl) [16]. The horizontal black lines separate clusters of cells with similar CNV events arranged in a hierarchical tree on the left. The horizontal red line shows the boundary between the CNV clusters of AdMer35 and AdMer43 cells. Track M shows CNV clusters with mixed cells from both cell lines. A large mixed cluster (*n* > 100) of AdMer35 and AdMer43 cells with similar CNV events highlighted by a light magenta rectangle and labeled “AdMer35/43” (enlarged in (**B**)).

**Figure 5 cells-15-00115-f005:**
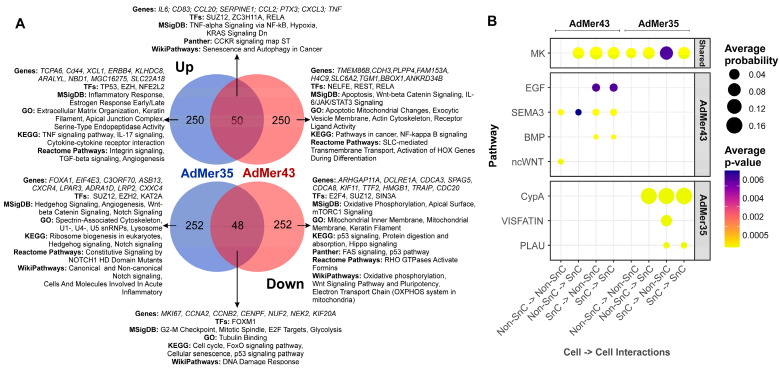
(**A**) Venn diagrams of top 300 up- or downregulated transcripts in SnCs compared to non-SnCs of the AdMer35 and AdMer43 cell lines. DEGs analysis of SnCs vs. non-SnCs was performed independently for AdMer35 and AdMer43 lines using Seurat FindMarkers with default parameters. Selected results of gene enrichment analysis are represented. (**B**) A bubble diagram of SnC intra- and intercellular interactions, constructed from CellChat V2 analysis of single-cell transcriptome data from AdMer35 and AdMer43 cell lines. Intercellular interaction classes for general and lineage-specific interactions are shown separately. Presented at the levels of intercellular interactions common (Shared) to both cell lines and specific to a particular line.

**Figure 6 cells-15-00115-f006:**
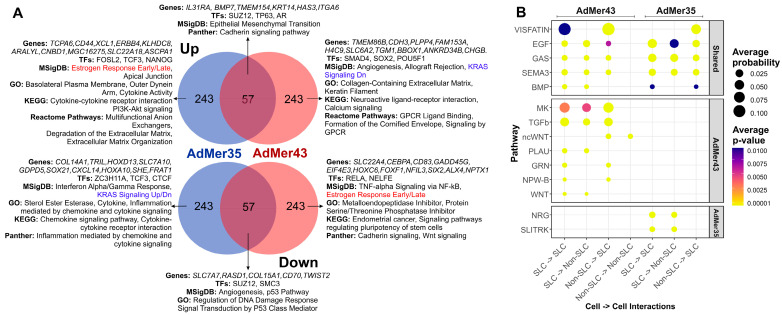
(**A**) Venn diagrams of top 300 up- or downregulated transcripts in SLCs compared to non-stem cells of the AdMer35 and AdMer43 cell lines. DEG analysis of SLCs vs. Non-SLCs was performed independently for AdMer35 and AdMer43 lines using Seurat FindMarkers with default parameters. Selected results of gene enrichment analysis are represented. Red and blue text highlights the switching of estrogen response gene expression and KRAS signaling between SLCs in cell lines. (**B**) A bubble diagram of SLC intra- and intercellular interactions, constructed from CellChat V2 analysis of single-cell transcriptome data from the cervical cancer cell lines AdMer35 and AdMer43. Intercellular interaction classes for general and lineage-specific interactions are shown separately. Presented at the levels of intercellular interactions common (Shared) to both cell lines and specific to a particular line.

**Figure 7 cells-15-00115-f007:**
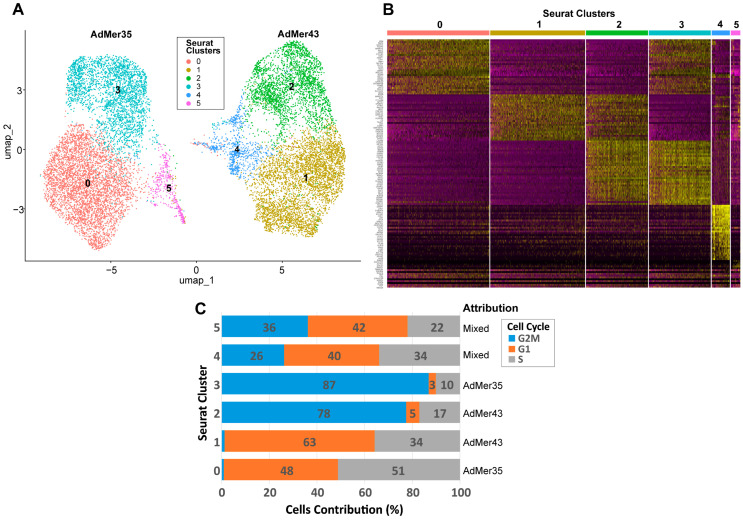
(**A**) UMAP plot of AdMer35 and AdMer43 cell classification by transcriptome clusters with Seurat function FindClusters (with resolution = 0.2 to search for the minimal number of clusters). (**B**) Heat map of top 30 upregulated transcripts from six Seurat clusters (upper line) detected in combined array of AdMer35 and AdMer43 single-cell sequencing data. (**C**) Distribution of cells in Seurat clusters by cell cycle phases and indication of the assignment of clusters to cell lines.

**Figure 8 cells-15-00115-f008:**
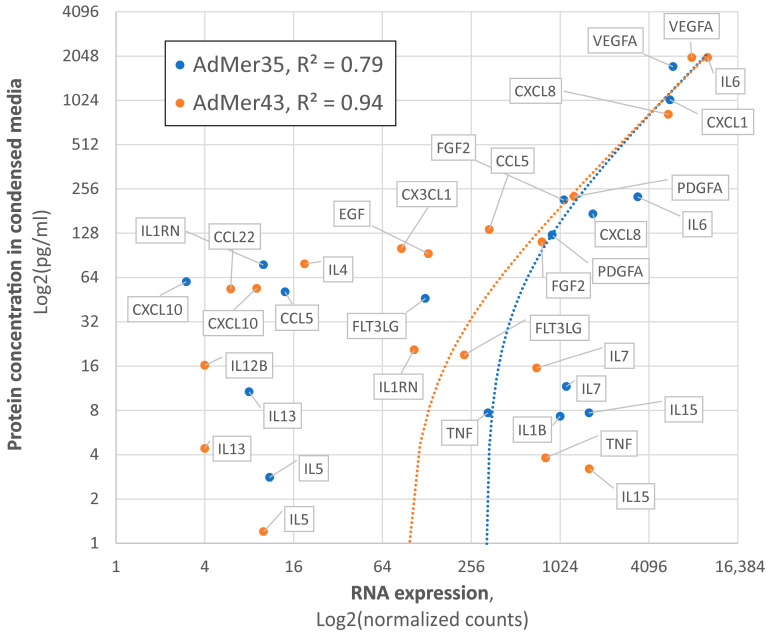
Correlation between single-cell RNA expression data and concentrations of secreted cytokines and chemokines in culture media AdMer35 (blue) and AdMer43 (orange). Protein concentrations were evaluated with Luminex-based MILLIPLEX MAP Human Cytokine/Chemokine Panel. Data represented in Log2 non-linear scales. The dotted lines show the trend lines in linear coordinates for both cell lines.

**Table 1 cells-15-00115-t001:** Characteristics of Cell Lines and Single-Cell Sequencing Data.

Cell Culture	Patient Therapy	Number of Passages of Cultured Cells	Number of Cells in SC Analysis *	Number of Processed Reads *
AdMer35	Three cycles of polychemotherapy according to the scheme paclitaxel combined with a carboplatin.	12	7256	183,166,343
AdMer43	Chemoradiation with a total focal dose of 50 Gy supplemented with cisplatin administration. Session of intracavitary brachytherapy with an SFD of 5 Gy, external boost to the cervical tumor with 20 Gy, and retroperitoneal lymph node conglomerates with 20 Gy. Gemcitabine therapy.	12	7198	295,237,133

* seeksoultools analysis data.

**Table 2 cells-15-00115-t002:** Cell counts and relative contributions of senescent-like cells (SnC) in a combined array of AdMer35 and AdMer43 cells.

Cell Types	PLC	Non-PLC	PLC (%)	Non-PLC (%)
SnC	107	1401	26.03	9.55
Non-SnC	304	13,273	73.97	90.45
Total	411	14,674	100	100

**Table 3 cells-15-00115-t003:** Cell numbers and relative contribution of SnCs to cell cycle phases in a combined array of AdMer35 and AdMer43 cells.

Cell Type	Cell Cycle Phase					
	G1	S	G2M	G1 (%)	S (%)	G2M (%)
SnC	703	483	322	47	32	21
Non-SnC	4709	4333	4535	35	32	32

**Table 4 cells-15-00115-t004:** The distribution of SLCs in combined array of AdMer35 and AdMer43 data was determined using AUCell with a set of RNA markers of cervical cancer stem cells.

Cell Types	AdMer43	AdMer35	AdMer43 (%)	AdMer35 (%)
SLC	508	1001	6.61	13.53
Non-SLC	7177	6399	93.39	86.47
Total	7685	7400	100	100

**Table 5 cells-15-00115-t005:** Optimized Seurat clusters describing cell cycle phase, PLC, SLC, SnC content in AdMer35 and AdMer43 cell lines. The data represents the percentage of the total number of cells in the cluster, excluding the “Cell Number” row.

Characteristic	Seurat Clusters					
	0	1	2	3	4	5
Cell Number	4422	4102	2682	2677	788	414
**Cell Lines**						
AdMer35, %	99.95	0.07	0.04	99.89	19.42	35.99
AdMer43, %	0.05	99.93	99.96	0.11	80.58	64.01
Cell Line Attribution	AdMer35	AdMer43	AdMer43	AdMer35	Mixed 35/43	Mixed 35/43
**HPV59 Viral Transcripts, %**						
Median HPV59 Transcript Contribution	0.061	0.074	0.075	0.071	0.109	0.001
**HPV59 Class**	Medium	Medium	Medium	Medium	High	Low
**Cell Cycle, %**						
G1	47.85	62.94	5.48	2.99	39.85	41.79
S	51.18	35.79	17.04	10.09	33.88	21.98
G2M	0.97	1.27	77.48	86.93	26.27	36.23
**Cell Cycle Phase**	G1/S	G1/S	G2M	G2M	G1/S > G2M	G1/S > G2M
**Progenitor-Like Cells (PLCs), %**						
PLC	0.81	1.39	1.27	0.45	1.52	62.80
**Enriched with PLCs (>10%)**	No	No	No	No	No	Yes
**Cancer Stem-Like Cells (SLCs), %**						
SLC	13.89	7.09	4.40	13.56	15.23	0.72
**Enriched with SLCs (>10%)**	Yes	No	No	Yes	Yes	No
**Senescence-Like Cells (SnCs), %**						
**SnCs**	10.33	11.38	5.03	7.88	16.12	26.81
**SnC Contribution**	Medium	Medium	Low	Low	High	High
**Short Description ***						
**Main Cell Line** Main Phase of Cell Cycle	**AdMer35** G1/S	**AdMer43** G1/S	**AdMer43** G2M	**AdMer35** G2M	**Mixed 35/43** Mixed	**Mixed 35/43** Mixed
**Enrichment**	SLC	No	No	No	SLC, HPV59, SnC	PLC, SnC (Lowest SLC, HPV59)

* Detailed in Appendix A.

## Data Availability

All data supporting the findings of this study are available in the article and Appendix A.

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
