# Peer review of "Single-Cell Transcriptomic Landscape of Cervical Cancer Cell Lines Before and After Chemoradiotherapy"

_cells, 2026, doi:10.3390/cells15020115_

Round 1
Reviewer 1 Report
Comments and Suggestions for Authors
This study examines cervical cancer cell transcriptomic signature before and after chemoradiotherapy.
Major concerns:
- Please provide patients' detailed information and why patients were chosen. Including disease history, alcohol and tobacco use history, BMI/BRI, gender, metabolic disease history, lipid panel, number of sexual partner, HPV status, etc.
- If cell flow cytometry was used, any cell marker changes?
- Please compare growth rate and invasive rate of these 2 cell types
- How long was the chemoradiotherapy?
- The citation is fine. Since the study focuses on the single cell
transcriptomic analysis of cervical cancer cell before and after
chemoradiotherapy, the study needs to show the patient's info for the
cells generated.
Reviewer 2 Report
Comments and Suggestions for Authors
This manuscript presents an extensive analysis of two isogenic cervical cancer cell lines derived from the same patient before and after chemoradiotherapy. The study addresses an important and timely problem, which is the mechanism of chemoradioresistance. The dataset is rich, the experimental workflow is well described and the work was surely submitted to the proper Special Issue. I also appreciate the well organized and presented tables in Supplementary Materials. However, the manuscript would benefit from some revisions, followed by addressing the comments presented below.
Detailed comments:
I appreciate the “highlights” section, but they should be organized differently, please check the instructions at https://www.mdpi.com/journal/cells/instructions
The study aims would benefit from a clearer, more explicit statement. I really miss the sentence like “The aim of this study was to ….”
Lines 55-58, The role of HPV 16 in the development of cervical cancer should be clearly stated, i.e. by citing https://doi.org/10.56782/pps.381
Please clarify why seeksoultools was chosen over more commonly used pipelines i.e. Cell Ranger?
Lines 179–181: Please indicate whether multiple-testing correction was applied consistently across all enrichment analyses and DEG selections.
Why this particular DEG cutoff has been chosen?
Line 475–480: While I agree that the identification of PLC based primarily on minimal CNV burden is possible, but alternative interpretations such as technical noise, partial normalization toward reference should be discussed.
Line 608–619: Please clarify whether these pathways are co-activated within the same cells or represent parallel signaling routes across subpopulations.
Figure 7, for AdMer43 and AdMer35 how many were in G1?
Figure 8, (mkg/ml) is a wrong unit
Reviewer 3 Report
Comments and Suggestions for Authors
This manuscript presents an extensive single-cell transcriptomic analysis of a unique pair of isogenic cervical cancer cell lines derived from the same patient before and after chemoradiotherapy. The study integrates scRNA-seq, CNV inference, senescence and stem-like state analysis, and ligand–receptor communication modeling.
However, the manuscript suffers from conceptual, methodological, and presentation weaknesses.
-Many conclusions imply functional causality (e.g., senescent cells “stimulate tumor growth”, stem-like cells “ensure chemoradioresistance”, specific ligands “drive resistance”), but no functional validation is provided. CellChat-based ligand–receptor predictions are hypothesis-generating, not evidence of active signaling. Claims regarding pro-tumorigenic effects of SnC–SLC interactions should be clearly framed as inferred associations, unless validated experimentally. Please tone down causal language throughout Results and Discussion, or otherwise provide targeted functional validation.
-Although the paired isogenic design is a strength, all conclusions are based on one patient. Inter-patient variability is not addressed. It is unclear which findings are generalizable versus patient-specific. Explicitly acknowledge this limitation and avoid broad generalizations about cervical cancer chemoradioresistance. Claims should be framed as case-specific mechanistic insights.
-Senescence and stem-like cell definitions are weakly validated. Identification of SnC and SLC relies exclusively on custom gene signatures and AUCell scoring. No canonical senescence markers (e.g., CDKN2A/p16, CDKN1A/p21, SASP validation at protein level) are functionally validated. Stem-like cells are inferred without orthogonal confirmation (e.g., sphere formation, drug resistance assays, or known CSC surface markers). Please provide justification for marker selection, include robustness checks (alternative gene sets), and temper conclusions where validation is lacking.
-InferCNV results are used to define “progenitor-like cells” and therapy-driven clonal selection. InferCNV provides relative expression-based CNV estimates, not direct genomic validation. In my opinion, the term “progenitor-like” is speculative and insufficiently supported. Rephrase to “cells with lower inferred CNV burden” and clearly distinguish inference from genomic validation.
-Key analytical parameters are missing or unclear: AUCell thresholds are not justified. Multiple testing correction is inconsistently described. Criteria for cluster resolution optimization are insufficiently explained. Provide clear descriptions of statistical corrections, and clustering decisions.
-The Results section is too long and repetitive, with frequent re-explanations of known biology and pathway functions. Condense Results by: Reducing background explanations, Moving descriptive pathway biology to Discussion, Summarizing repetitive DEG interpretations
-Numerous grammatical errors and awkward phrasing reduce readability. Several sentences are overly long and should be simplified.
-Some figures (e.g., network diagrams) are overly complex. Improve the description of the figure legends.
- Avoid mixing “chemoresistance”, “radioresistance”, and “chemoradioresistance” without clarification.
-Some claims (e.g., microRNA functional roles) are speculative and need stronger or more cautious citation.
-HPV59 transcript differences are noted but not sufficiently discussed in terms of biological relevance.
Comments on the Quality of English LanguageModerate editing
Round 2
Reviewer 1 Report
Comments and Suggestions for Authors
No more comments
Reviewer 2 Report
Comments and Suggestions for Authors
The authors have revised and improved their work, this version can be accepted.
Reviewer 3 Report
Comments and Suggestions for Authors
The authors improved the manuscript a lot.